# Metabolic Food Waste as Food Insecurity Factor—Causes and Preventions

**DOI:** 10.3390/foods11152179

**Published:** 2022-07-22

**Authors:** Ioana Mihaela Balan, Emanuela Diana Gherman, Ioan Brad, Remus Gherman, Adina Horablaga, Teodor Ioan Trasca

**Affiliations:** 1Faculty of Management and Rural Tourism, Banat’s University of Agricultural Sciences and Veterinary Medicine “King Michael I of Romania” from Timisoara, 300645 Timișoara, Romania; ema.gherman@usab-tm.ro (E.D.G.); ioanbrad@usab-tm.ro (I.B.); remusgherman@usab-tm.ro (R.G.); 2Faculty of Agriculture, Banat’s University of Agricultural Sciences and Veterinary Medicine “King Michael I of Romania” from Timisoara, 300645 Timișoara, Romania; adinahorablaga@usab-tm.ro; 3Faculty of Food Engineering, Banat’s University of Agricultural Sciences and Veterinary Medicine “King Michael I of Romania” from Timisoara, 300645 Timișoara, Romania

**Keywords:** food security, sustainable and healthy food choices/systems, *Metabolic Food Waste* MFW, nutrition, overconsumption

## Abstract

The *Metabolic Food Waste* MFW (kg of food) first developed in 2016 as a new indicator by Serafini and Toti, indicates the amount of food consumed above the nutritional requirements, and the impact of this overconsumption on the environment. It is necessary to identify the causes and to develop potential methods to prevent and reduce MFW, at the same time as increasing consumer awareness about unsustainable diets and changing diet habits towards more environmentally conscious consumption patterns. This study was conducted by collecting and analysing existing reports and studies regarding nutritional requirements, consumer behaviour related to food consumption and food waste, environmental impacts of food waste, and the concept of Metabolic Food Waste. The process of data collection involved searching the studies available online, using keywords related to the subject of MFW and overconsumption. The references in the initial studies consulted were also analysed in order to further identify new data relevant to overconsumption and MFW. The materials studied and analysed related to the environmental impact of MFW were published by E. Toti and M. Serafini in 2016 and 2019; additionally, in order to understand the causes of overconsumption numerous studies were reviewed regarding consumer behaviour, the relationship between economic development and overconsumption, mental health and dietary habits, physical context and dietary habits, genetic predisposition, also childhood and early adulthood environment. By analysing and corroborating external data available for food waste, nutritional requirements, and the environmental impact of food waste and consumer behaviour, we identified as primary causes for MFW the lack of nutritional education and little understanding of the nutritional requirements amongst all categories of consumers, poor access to appropriate food resources or reduced availability of fresh produced food. We conclude that for the quantification of the negative impact of MFW on both the environment and human health we need decisive action to raise consumer awareness for healthy and sustainable diets, together with a uniform worldwide distribution system for nutritious food.

## 1. Introduction

The idea of overconsumption as a means of wasting food gained traction in the early 2000s, when Vaclav Smil from the University of Manitoba highlighted the growing gap between food production and consumption, mainly in developing countries, mentioning an increase in calorie intake from 1000 kcal/day to 1500 kcal/day in high-income countries in the last decades. He states that this increase should be considered as a wasteful food habit. Since then, the subject of overconsumption has been discussed in literature, being mentioned by the Barilla Center for Food & Nutrition (BCFN) and by Serafini and Toti when they introduced the term *overconsumption* to the concept of MFW. MFW_(kg of food)_ is an indicator developed in order to express the quantity of food consumed above the nutritional needs of the human body, resulting in the accumulation of excess fat. The indicator can be further used to determine the impact of overconsumption on the environment, expressed in carbon, water and land footprint as MFW_(kgCO_2_eq)_, MFW_(×10 L)_, and MFW_(×10 m^2^)_, respectively. The current research aims to identify the causes of overconsumption, as food waste is a major issue in trying to ensure global food security, and wasteful behaviours need to be understood in order to address them. We believe that the main reason for overconsumption and obesity is the lack of nutritional education among the consumers. Informing consumers about their physiological needs regarding food intake and educating them in choosing healthy and sustainable food products can have a major impact on reducing the occurrence of food waste in the form of excessive consumption. The main objective of the study is to identify at least one method of prevention when it comes to MFW, together with an outline of the reasons why it occurs [1,2,3].

## 2. Study Objectives

In order to understand the multitude of aspects of overconsumption and the relationship between overeating and food waste, we made a comprehensive review of the literature. The authors consulted online studies related to overeating and food waste, using a combination of keyword searches such as: “overconsumption + food waste”, “overconsumption + causes”, “overeating”, “overeating + food waste”, “overeating + causes”, “overeating + prevention”, “food waste”, “household food waste”, “consumer food waste”, “obesity + causes”, “obesity + prevention”, “metabolic food waste”, “nutritional guidelines”, “nutritional needs”, “nutritional requirements”, and “adequate nutrition”. We also consulted the references in each study examined in order to identify new data regarding MFW, overconsumption of food and potential causes and prevention measures. The materials studied and analysed related to the environmental impact of MFW were discussed in the studies published by E. Toti and M. Serafini in 2016 and 2019 [3,4]; key sources reviewed covered consumer behaviour [5,6,7,8,9,10,11,12,13,14], the relationship between economic development and overconsumption [12,15], mental health and dietary habits [16,17,18,19], physical context and dietary habits [20,21,22,23], genetic predisposition, and the childhood and early adulthood environment [11,23,24,25].

Understanding an adult’s nutritional needs, in relation to gender and intensity of activity, is essential when trying to assess what can be considered as excess nutrients that can lead to overweight or obesity. In this regard, Fischer and Garnett‘s *Plates*, *pyramids*, *planet* (2016) and the Barilla’s *Double Pyramid* (2016) both address the issue of recommended diet, overconsumption, and their impact on sustainability. Therefore, the recommendation is to consume a precise meal structure [26,27] (Figure 1):-Extras, such as sweet, fatty snacks, alcohol (one serving);-Fat and oils (two servings);-Dairy products (three servings) and meat, fish or eggs (one serving);-Read, cereals, side dishes (four servings) and vegetables and fruits (five servings);-Beverages (six servings).

We consulted studies and reports from the last decade in order to analyse: the correlation between overconsumption and other food wasting behaviours; the level of knowledge and awareness of healthy and sustainable eating habits; how to determine a possible relationship between food waste behaviours and eating habits; and existing data on consumer behaviours related to food purchases (checking supplies before shopping, making a list, following the list, proper storage and preparation etc.). This research enables us to assess the presence of responsible consumption habits related to consumption and food waste, and the comparative analysis of the previously presented data was used to conduct this study [1,2,5,6,7,8,9,10,26,28].

The objective of the present review is to identify external and internal factors that lead to overeating associated with MFW, and some potential measures for preventing or correcting this behaviour.

## 3. Discussion

Human nutrition has changed drastically in the twentieth century and since. From local and seasonal food, the developed world has made the slow transition to a global approach to food and the availability of fresh produce throughout the year. Nowadays, exotic fruits and vegetables are readily available on every supermarket shelf in the developed world, and the food industry has discovered ways to preserve all foods to extend their validity and availability. Not only that, but the availability of high-quality products such as meat and diary increased exponentially over the last century, along with an increase in basic income and education. All of these factors and many others have led to a change in the human diet, which translates into an increase in the calories consumed daily, mainly due to an increase in the consumption of animal protein from meat, dairy and eggs. Data from the Food and Agriculture Organization of the United Nations, from the 2000 Food Balance Sheet show that food availability at retail level is over 3000 kcal/day/capita, the highest values being registered in USA—3600 kcal/day/capita, EU—3500 kcal/day/capita and Canada—3300 kcal/day/capita. Contrasting with this information, for an urban, sedentary, old population (characteristics of the population in the northern hemisphere) requires between 1500–2000 kcal/day/capita for female adults and 2000–26,000 kcal/day/capita for male adults. Therefore, in the developed areas of the globe, the availability exceeds real nutritional needs by approximately 1000 kcal/day/capita, and even more in USA—1500 kcal/day/capita. Accounting for unavoidable food loss and with ensuring a safety margin, the average supply per capita is considered to be 30% higher than average need, resulting in an average need of 2600 kcal/day/capita. The difference between need and availability of 700 kcal/day/capita registered in the developed countries is either food waste or overconsumption, resulting in metabolic food waste [29].

Jobs becoming more and more static, a decrease in the level of physical activity during the day and increased availability of food led over time to what is now called “the obesity pandemic”.

According to data from 2019, 35.20% of the USA population is overweight, and European Union statistics show that in the same year, 52.7% of its population was overweight. Globally, the prevalence of obesity tripled between 1975 and 2016, according to data provided by World Health Organization WHO. These facts raise many questions, not only about to the health sector and the economic burden of preventing and treating diseases related to overweight and obesity, but also about the ethics of overconsumption [11,30,31,32,33].

Is it moral and ethical to consume food in excess, knowing the related health and financial burden, while exacerbating the burden on the planetary production system? What are the factors that determine the adoption of an unhealthy diet? What can and should be done to reduce the occurrence of overconsumption in the developed world?

### 3.1. Overconsumption as Food Waste

The first scientist to raise the issue of food overconsumption as a waste-eating behaviour was Vaclav Smil (2004). He argued that food production can be optimized not by increasing physical inputs to boost production, but by reducing inefficiencies, post-harvest losses and by “matching more closely actual food needs and availability”. He further stated that the level of waste occurring in the food system is one of “the most offensive demonstrations of human irrationality”. Smil focused on the growing gap between food production and consumption. Although North America and Europe are leaders in wasteful behaviours and overconsumption, these are spreading rapidly among the upper classes in middle- and low-income countries due to increasing incomes and the availability of high-caloric foods such as meat, dairy and eggs. The BCFN report from 2012 also emphasizes the idea that overconsumption is a wasteful behaviour and should be addressed in the global fight against food waste. Serafini and Toti went further and developed a new indicator—*Metabolic Food Waste* MFW—expressed in kg of food, which corresponds to the excess of food consumed over physiological needs, leading to overweight and obesity. It is not only human health that is affected by overconsumption; the MFW_(kg of food)_ being used to determine the environmental impact of overconsumption also expresses in carbon—MFW_(kgCO_2_eq)_, water—MFW_(×10 L)_, and land footprint—MFW_(×10 m^2^)_ [1,2,3].

The complete information from 86 countries was analysed by Toti and others in 2017, and these countries were classified in the seven regions of the FAO world. According to them, the global impact of MFW in the world corresponds to 140.7 million tons of food waste associated with overweight and obesity. It was noted that of all the regions analysed, Europe was responsible for the largest amount of MFW (39.2 million tones), followed by North America and Oceania (32.5 million tones) and Latin America (20 million tones), while the lowest amount of MFW was recorded in Sub-Saharan Africa, with only 5 million tones [4] (Table 1).

It is noteworthy that cumulating the quantities of MFW in Europe and North America and Oceania, the amount is over 50% of the total MFW worldwide, reported in 2017, at the level of 86 countries analysed (Figure 2).

MFW can be further developed to express the impact of overconsumption on the environment, such as carbon, water, and land footprint. Toti and others argue that excessive consumption of energy-dense foods, such as meat and animal fat, fried foods, dairy products and sweets, represents a significant environmental cost, being the main contributor (approximately 80%) to the increase in GHGs from food production. On the other hand, a balanced diet, based mainly on vegetable foods, is not only a healthier option, but has low risks for the diet–environment–health triangle. BCFN also states in their report that, as consumption in developed countries significantly exceeds the recommended caloric needs, the phenomenon of overeating should be taken into consideration in future discussion about food waste. The Lancet Commission advocates responsible consumers behaviour, encouraging a reduction in animal foods and a more balanced diet, for both environmental but and human health [2,4,34].

### 3.2. Causes of Overconsumption

When we refer to food, overconsumption means overeating, and is the situation in which an individual consumes food above the body’s energy requirements in relation to energy expenditure, leading to excess fat in the body. When practiced constantly for long periods of time, and coupled with lack of physical activity, overeating is the principal cause of overweight and obesity in both adults and children. Another aspect of overconsumption is that it is mainly due to an excessive intake of free sugars, fat, animal products and alcohol [35,36].

The factors that lead to overeating are multiple, and difficult to identify exactly, but they can be grouped in helpful categories: global factors, societal factors, educational factors, and individual factors.

#### 3.2.1. Global Factors Leading to Overconsumption

Among the most important factors that have led to an increase in the amount of food globally is the economic growth experienced in recent decades by economies around the world. Economic growth has been manifested by the increased availability of food resources in the northern hemisphere, due to innovation in agricultural practices that translate into higher quantities of food available for consumption, correlated with a decrease in the price of products previously considered high-end products (meat, dairy). Economic growth has also manifested itself as an intensification of global trade practices, further contributing to increased availability in previously middle- and low-income countries. Economic growth also leads to an increase in per capita income, due to the opening of borders for foreign investors who expand the number of jobs available within the country’s borders. Increased availability and income have led over the years to an increase in the amount of food people eat, leading to overconsumption and therefore metabolic food waste [12,15]. Although contrary, food waste and food insecurity can coexist in the same area, even in the developed economies of the globe. Overconsumption and undernutrition coexist among the population due to multiple factors: on one side, an emergent middle class, population transition towards urbanized areas and the increase in income determine dietary changes characterized by an increase in consumption, while on the other side perpetual inequalities start to manifest from early ages: reduced access or impossibility to access quality food resulting in delayed development, difficult physical or economic access to quality education determining later in life a reduced access to high-paying jobs, thus perpetuating the cycle of insecurity. On the other hand, high access and affordability reflect in all areas of life, therefore continuing the cycle of abundance, and in many cases encouraging overconsumption [20].

#### 3.2.2. Societal Factors Leading to Overconsumption

The most important driver of overconsumption, when it comes to societal factors, is the current trend of consumerism. Since the beginning of the twentieth century and during the beginning of the mass production of goods, consumerism has spread rapidly due to aggressive marketing campaigns. It is characterized by the purchase of goods and services in an ever-increasing quantity and the rapid withdrawal of goods that are still functional in favour of the “new”. In terms of food purchases and consumption, it encourages a display of wealth through large quantities of food purchased and consumed over energy requirements, promoted by marketing strategies that often end as food waste, either by throwing away leftovers, or by eating over the physical needs [37,38].

Another societal factor is the habits of food consumption that are specific to every culture in the world. Some cultures have traditions that encourage the consumption of high-calorie food, especially during the holidays, when full meals and full bellies are signs of joy, happiness and wealth.

Another factor leading to overconsumption is the expansion of fast prepared food in all middle- to high-income countries. The issue with this type of food venue is that most often the foods sold are high in sugar, salt and fat. All of these components are correlated to a high level of satisfaction, leading to repeated consumption and overconsumption. They also lack the micro and macronutrients needed for normal physical function, so in addition to favouring overconsumption and leading to overweight and obesity, if eaten frequently over a long period of time they contribute to a deficiency of nutrients [2,21].

Food waste and nutrition are closely linked, as waste, even as MFW determines a reduction in global availability. Food waste not only affects environmental health due to the unsustainable use of resources for food production, but it also affects human health in multiple aspects: reduced availability affects those with reduced means for purchase, and determining undernutrition while wasting food by overconsuming directly affects the health of individuals that regularly consume more than necessary. At the level of society, both of these issues are a matter of sustainability and equity of the food chain [39].

#### 3.2.3. Individual Factors

Individual factors leading to overconsumption are linked to the eventual consumers, their food choices, and their dietary patterns. This category of factors includes the elements of mental health that can affect food consumption patterns: excessive stress can lead to stress-eating behaviours resulting in overeating; and strong emotional periods that can disturb the feeding pattern can also leading to overconsumption. In the case of stressful and emotional eating, the root cause to be addressed is the mental health of the individual (and the ways to deal with emotionally challenging events), not eating behaviour—it is just an external manifestation of internal turmoil. For this reason, we will not focus on this paper on the issue of stressful behaviour and emotional overeating [2,16,17].

Other factors that lead to food waste, overconsumption and subsequently to metabolic food waste, which are directly dependent on the individual [13] and will be further addressed, are considered to be: improper storage conditions and cooking methods in households [40,41]; poor management of food stocks inside households [42]; the level of food security or insecurity experienced by the members of the households [43]; lack of education and knowledge regarding individual energy needs; the desire to buy and eat associated with a lack of awareness when eating food—eating while walking, working, watching TV; lack of control over food choices—people who are dependent on someone else (the elderly, women, children, young adults); personal preferences of food groups; and attitudes towards food waste—negative emotional responses (guilt, sadness etc.) when throwing food away; and not being aware that overconsumption is also food waste [2,3,40,41,42,43,44].

Although food waste continues to be a global issue that greatly affects the sustainability of the food chain and planetary health, it is slightly different from metabolic food waste, which also directly impacts the individual health of people. Food waste can be generated in the households by improper storage conditions, which continue to represent an issue in the developed world. Even with the great development of technology allowing for food storage in controlled environments (chilled storage), improper use of the technology inevitably results in waste. A study from 2020 on German consumers determined that less than 50% of the consumers respect the recommended temperature settings of 4 °C in their refrigerators, leading to 30–50% waste of fresh fruits and vegetables, 15–35% waste of prepared foods, and 4% waste of fresh meat and fish [40]. Furthermore, poor home economics skills and poor management skills for food stocks inside the household contribute to an increase in food waste at household level. A study from 2021 conducted by researchers at Poznań University of Life Sciences on students and lecturers at the University revealed that in younger persons’ households, where they are starting life on their own, the level of food waste tends to be higher than in households where older persons are responsible for food commodities management. A greater number of students (almost 90% of respondents), compared with employers (less than 50% of respondents), reported that although they check their home stock before shopping, they buy foods based on a shopping list but they also tend to buy food products that are not on the list under the influence of momentary impulses: cravings, shopping while hungry, discounts etc. [42]. The individual level of food security is also an important determinant of the level of food waste, as shown by a nationwide cross-sectional study in Saudi Arabia. Due to limited agricultural production determined by geography and climate, Saudi Arabia imports around 80% of the food needed, with an expected increase in the trend in the coming years. Even at this level of dependence on external production, a high level of food waste is still registered. From the participants of the study, 13.3% were part of severely food insecure households. For the food insecure household, there was a difference in fresh foods wastage, with higher levels registered in households that benefited from social aids. For the cooked food waste, higher levels were registered in households that were food secure than in those living with food insecurity. The study revealed that income, including social aids, and the presence of elderly members in the household and the number of members of the household were all determinants of both the level of security or insecurity and the level of food waste, cooked or uncooked [43].

Although food waste at the consumers’ end represents 61% of the total amount of food waste and loss in North America and Oceania and 52% across Europe, the discussion thus far has omitted to include overconsumption of food products as a wasting behaviour.

#### 3.2.4. Lack of Education and Knowledge Regarding Energy Needs

In recent decades, the Western world has been bombarded with information about so-called healthy diets, which promise not only to get rid of overweight individuals, but also to improve their long-term conditions. This fad diet roller coaster has had exactly the opposite results than the desired ones. Knowing that any diet has the reverse, to gain weight and then something extra after it is over, individuals are now in the position to binge diet, following one restrictive diet after another, for fear of not regaining the weight they just lost. Another thing to note is that most of the time, these diets revolve around restriction (Keto diet—restrictive carbohydrates; Paleo diet—dairy, legumes, cereals; low calorie diets; diet elimination food groups; restrictions of food combinations etc.), and the body’s long-term depletion of the basic macro- and micro-nutrients needed for optimal functioning. Individuals lack the basic knowledge and understanding of the principal functions of their bodies, as well as the nutritional requirements to ensure optimal intake, and so they choose to follow a diet for a short time without being aware that in fact, a healthy diet is not a diet at all but rather a way of life.

In general, individuals overestimate the amount of protein and fat needed and underestimate the importance of vitamins and minerals found in fruits and vegetables. The result is that they eat far more animal products than the energy requirements and far fewer fresh fruits and vegetables. A study from The Lancet Commission points out that globally, red meat intake is twice the recommended intake, with North America exceeding the recommended intake six times and Europe and Central Asia four times. The same study shows that, compared to the recommended quantity of fruits and vegetables consumed, North America consumes only 60% of each category, and Europe and Central Asia consume about 55% of the recommended consumption of fruits and 70% of the recommended vegetables. Both North America and Europe and Central Asia consume more than the recommended amount of red meat, starchy vegetables, eggs, poultry, and dairy, but less than the recommended fish, vegetables, legumes, whole grains and nuts. These imbalances in the intake are in line with an unhealthy lifestyle, being clear indicators that the population either does not have the knowledge or chooses not to follow the recommendations of the authorities regarding nutrition [34].

#### 3.2.5. The Impulse to Buy and Eat Related with Lack of Awareness When Eating food—Eating While Walking, Working, Watching TV

Derived from the trend of consumerism, the issue of impulse buying is consistent with behaviours such as overbought and overeating. This behaviour mostly manifests either in response to external factors—promotions, discount prices, product display, store layout, other marketing strategies that encourage consumers to buy more than necessary, or in response to internal (personal) factors—predisposed to be influenced by marketing strategies, inability to withstand momentary urges (determined by seeing, smelling products), lack of preparation before and during shopping (plan meals in advance, determine the amount needed for each food product, check existent stock at home, make a shopping list, include possible “triggers” in the list, follow the list). Closely related to impulsive buying is impulsive eating. While unrelated to mental health issues, as discussed above, impulsive eating manifests itself as an inability to resist the need to eat, even if you are not hungry at that time. Impulsive eating behaviours are constantly checking the fridge, eating foods just because they are “there” and triggering the need to eat, inability to resist another bite even when you are already full and satisfied, eating because you see someone else eat and are unable to resist the impulse. Impulsive eating should not be confused with compulsive eating, which is represented by repeated episodes of overeating, even to the point where it creates physical discomfort, but also a constant concern for eating and eating. Compulsive eating is considered an eating disorder and should be addressed by a professional, to guide and aid the individual suffering [18,19].

Closely related to impulsive eating is the lack of awareness and mindfulness directed towards the process of food consumption. From family and even community activity, food has become more and more an individual activity. The time allocated for this activity has also changed. If in the past meals were a reason for the family or community to get together and spend time together, today the meals are often taken alone and in a hurry. The busy schedules and the various other activities that fill our days make eating in a dedicated time slot almost impossible. Most of the time, the meal is cramped between meetings and deadlines or is consumed in a hurry, while going from one place to another; therefore, mindful eating becomes more and more a treat. Another thing worth mentioning is that even if we take time for a meal modern society likes to eat while watching TV, an activity that exhausts the meal and the attention it deserves. Nutritionists strongly claim that the lack of awareness and mindfulness when eating is one of the main causes of overeating. Not paying attention to the food, and to the signals that our body sends us, correlate with eating too fast that inevitably leads to overconsumption. This behaviour, if constantly repeated, will lead to overweight and obesity and related health issues [1,2,26,27,41,42].

#### 3.2.6. Personal Preferences, Allergies, Intolerance

It is known that everyone is unique, and food preferences are no exception. Each of us has personal tastes and tends to like some foods more than others. Others suffer from food allergies and intolerances, making it impossible to eat certain food. This, in turn, makes the effort to create a single diet for all to be futile [26,41].

Food preferences can lead to overconsumption of certain foods, such as meat, dairy, fats, and carbohydrates, which trigger a higher reward response, to the detriment of other foods such as fruits and vegetables, which are healthier and more beneficial for us but do not trigger the same neural reward. Another aspect to consider is that when we eat foods that we like, we tend to eat more than we need to in order to ensure our sense of satisfaction. As with any other trigger of the reward system, the more and more often we eat what we like, the more we want to eat it, which generally leads to long-term overconsumption, leading to overweight and obesity. Food preferences can be recognized and controlled by changing the pattern of consumption to a healthier one, which includes all food categories in the recommended amount for each individual [18,26].

People who are affected by allergies and intolerance are guided to replace foods that pose a threat with appropriate foods, in terms of their micro- and macro-nutrient content, to ensure adequate dietary intake. People suffering from allergies and intolerance are not advised to abuse a food group to replace the energy needs corresponding to the food they cannot eat. For example, someone may be allergic to nuts, but this does not mean that they should consume an excess of fish to ensure the intake of Omega-3 fatty acids, as this element is also found in vegetable oils, flax seeds, chia seeds and leafy vegetables [26,34,41].

#### 3.2.7. Lack of Control over Food Choices—People Who Are Dependent on Someone Else (Elderly, Children, Young Adults, People with Disabilities)

Overconsumption of food can be an effect of the personal situation, as is the case with children and young adults, the elderly, people with disabilities and some women who have no control over the food they eat.

In the case of children and young adults, the elderly in the care of others and the disabled who need constant care, they are in the situation to eat what the person in charge of their care provides or cooks for them. If the person does not have the knowledge related to the energy needs of a child (in particular) or of the individual in general, it can easily lead to all forms of malnutrition, including over-nutrition. This is the case for many overweight and obese children who are fed an inadequate diet or are asked by their supervisor to finish everything on the plate, even if it is too much for them. In adulthood, this behaviour of finishing food on the plate will persist, even without supervision, and will eventually be practiced with their own children and creating a continuous chain of over-nutrition. In the case of the elderly, they may be overfed due to concerns about their general health. The person in care of the elderly may not realize that the energy requirements of the elderly are lower than those of other adults, due to changes in the body but also to decreased physical activity. Therefore, the elderly may be asked to eat more than they need, with good intentions—a concern for their health—that leads to overweight and obesity, which are even more problematic for geriatrics [11,22,23].

#### 3.2.8. Attitudes towards Food Waste

The way individuals relate to food waste differs greatly, from a total disregard for the negative aspects of food waste to an almost compulsive fear of it. It should also be borne in mind that overconsumption as food waste is a new concept and still unknown to consumers. Until recently, the only problems with overconsumption were the health risks of being overweight and obese. Moral attitudes toward food waste, such as guilt or remorse for throwing away waste, often lead to overconsumption. Individuals do not yet realize that eating over the required amount is a waste of resources, which has a negative impact on personal health. There are many ways to deal with leftovers—from better meal or portion planning to reusing leftovers to creating new meals or even preserving them safely for later consumption—that do not involve excessive consumption [5,7,9,13,14].

### 3.3. Methods to Prevent Overconsumption

Prevention of metabolic food waste is the most effective way to address both the problem and its effects—health issues, environmental issues, moral issues. Unfortunately, universally valid ways to prevent people from engaging in irrational overconsumption practices have not yet been found. With the level of complexity that our current society presents, it is impossible to address every factor that underlies the metabolic food waste. We must not forget that overconsumption is a problem specific to the middle- to high-income strata of society, while individuals who are currently in the lower strata are still affected by insufficient food resources available or the inability to access these resources due to limited or non-existent trading opportunities for those resources.

Another aspect of consumption to consider is that economic theory holds that consumers continue to act in an irrational manner, making decisions based on feelings, beliefs, and heuristics, not considering the facts presented, but the feelings they experience when buying or consuming a product. This is the case with food technologies, for example, when consumers who faced food designed by humans, such as genetically modified products, edible films with nanotechnology coating or meat grown from stem cells, perceive it as unnatural as disgusting. The same flawed belief system applies to foods marketed as healthy: the consumer strongly beliefs that healthy foods cannot be tasty, although there are not scientific studies to prove that indeed healthy food cannot be tasty [45,46].

In view of all the above reasons, we believe that preventive actions should be strongly consumer oriented. As mentioned in the introductory section, we want to identify at least one possible method of preventing MFW that can be applied globally, regardless of geographical location, food system structure or diet choice. Focusing on consumer behaviour, we assume that the greatest impact on reducing the appearance of the world MFW will be the education of consumers around the world.

Nutritional information on the energy intake required for the optimal function of the body system should be available from the earliest stages of life, and rational intake behaviours should be encouraged and rewarded from childhood. It is known that in the early years memory functions like a sponge, absorbing every piece of information it receives. It is also known that strongly developed and established early behaviours tend to remain stable in adulthood. Therefore, we hypothesize that children’s nutrition education, targeted at each stage of development, has a strong potential to change current consuming behaviours. Educating children also has great potential to indirectly educate the adults around them. Families tend to evolve together and members influence each other, so the educated child is more likely to influence their parents in making better dietary decisions [24,25].

From an adult perspective, it is more difficult to change flawed behaviours and beliefs, but in political theory and behavioural sciences, nudging or choice architecture, a concept popularized in 2008, has been proven to increase the likelihood of positive behavioural change and promote the adoption of a healthier way of life. It has been shown that nudging has a greater impact on choice than any restrictive methods, based on imposition, coercion, banner, or corrective actions. An example of pushing the consumer to choose a healthier option (for example fresh fruits) is to place them at eye level. As a method that has been shown to work in previous studies, we hypothesize that actions aligned with global consumer education, combined with nudging practices to guide the choice of a healthier option, could have a major positive impact on reducing the incidence and prevalence of MFW globally. We believe that continuous education in food and nutrition is of great importance, as knowledge of the facts and gentle guidance together is the best strategy for orienting consumers towards a healthy alternative life. As there are no external influences that would lead the consumer to overcome their prejudices, this is seen as an internal effort. However, knowing about biases and how to overcome them is a first step in empowering the consumer to make rational, fact-based decisions rather than irrational biased decisions born of misconceptions [13,47,48].

We therefore hypothesize that the best way to prevent MFW and its effects is to start nutrition education in the earliest stages of development, associated with adults nudging practices, where necessary.

## 4. Conclusions

This present paper aims to identify the potential causes of MFW and to formulate a hypothesis on the prevention of MFW. We focused mainly on the causes related to consumer behaviour and highlighted the most common:Global factors—economic growth, intensification of global trade practices, increased availability, increased income per capita;Societal factors—consumerism, food culture, expansion of fast prepared food in all middle- to high-income countries;Individual factors—food choices and preferences, dietary patterns, mental health, lack of education and knowledge regarding individual energy needs, lack of awareness when eating, lack of control over food choices, attitudes towards food waste.

The list of causes is by no means exhaustive, as a complete list of factors leading to overconsumption would require extensive, globally applied studies without guaranteeing that the causes are exhausted, and so the causes presented in this paper cover a wide range of MFW.

As a possible preventative approach to the problem, we focused on early childhood education, with great potential to influence the behaviour of their responsible adults, and the niche architecture of choice associated with lifelong education, addressing both nutritional and rational thinking as a method of guiding adults to the behavioural changes needed to reduce the incidence and prevalence of MFW and its effects.

Informing adult consumers about their physiological food and educating them in choosing healthy and sustainable food products, consumed in adequate quantity, can have a major impact in reducing the occurrence of food waste in the form of excessive consumption.

## Figures and Tables

**Figure 1 foods-11-02179-f001:**
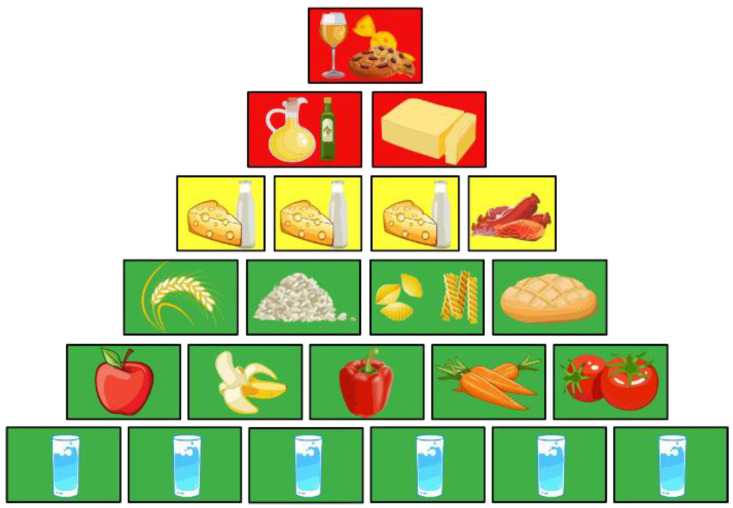
Recommended dietary intake by United Nations *Plates*, *pyramids and planets* based on an idea by S. Mannhardt—authors adaption after [27].

**Figure 2 foods-11-02179-f002:**
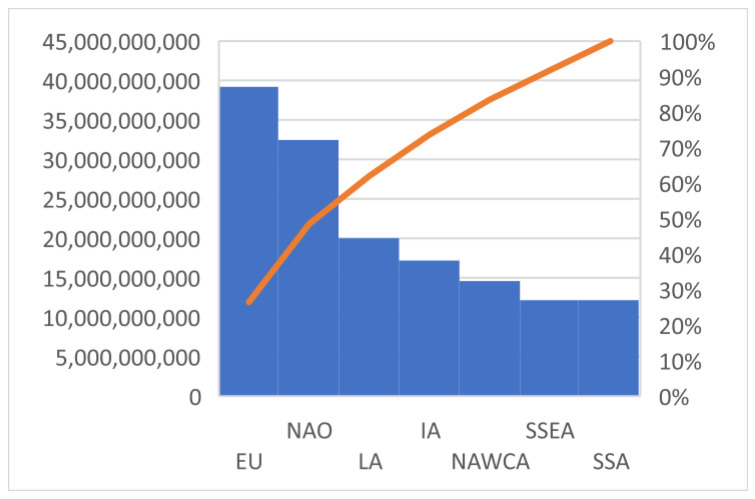
Worldwide distribution of MFW (to of food) associated with overweight and obesity.

**Table 1 foods-11-02179-t001:** Metabolic Food Waste MFW (tons of food) associated with overweight and obesity—authors’ selection after [4].

World Region	MFW (to of Food)
Europe (EU)	39,201,410.847
North America and Oceania (NAO)	32,465,755.707
Latin America (LA)	20,022,343.875
Industrialized Asia (IA)	17,190,412.965
North Africa, West and Central Asia (NAWCA)	14,595,049.642
South and Southeast Asia	12,181,476.616
Sub-Saharan Africa (SSA)	5,079,066.441
**TOTAL WORLDWIDE**	**140,735,516.093**

## Data Availability

Not applicable.

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
