# Peer review of "Metabolic Food Waste as Food Insecurity Factor—Causes and Preventions"

_foods, 2022, doi:10.3390/foods11152179_

Round 1
Reviewer 1 Report
Thanks to the authors for providing this useful study.
This study provides valuable useful information on “Metabolic Food Waste as Food Insecurity Factor – Causes and 2 Preventions”. However, please pay attention to the following notes:
In Abstract:
· It should be indicated to the method of work used in the abstract, after the objective of the study, i.e. in line 14 approximately in this study.
Keywords:
· Include the term The Metabolic Food Waste [MFW] in the study keywords.
In introduction:
· Expand the introduction more by expanding more in the explanation of the concept of MFW and also inferring from recent references on the topic from several different countries around the world.
· What was mentioned in lines 36-39 must be conveyed and used in conclusion or discussion (if any).
In Materials and Methods:
· Research methods are very succinct.
· Expand more Explanation of the study methods according to the ideas put forward in the results sequentially.
· Include the publication date of both references: Fischer’s and Garnett’s “Plates, pyramids, 47 planet” and the Barilla’s “Double Pyramid” in the body of the text on lines 47 and 48.
In Results:
· Reconsider the way the references are placed in the end of the paragraphs. It means placing the references directly at the end of the reference idea and not stacking the references at the end of the paragraph (for example: lines 66 & 87.
· Line 116: The title of Table 1 does not match the content. Since it contains only Metabolic Food Waste but overweight and obesity are shown in Figure 1.
· Line 198: Revise paragraph 3.2. 4. Are all of what is mentioned in it the authorship of the authors of this study? Because there is only one reference at the end of the paragraph in line 226. Or that everything mentioned in the paragraph was taken from the same reference, and this (if it is) needs to be reviewed and other supporting references added.
· The study focuses on the excessive amount of Metabolic Food Waste (MFW), but it is better to mention the quantities of food wasted due to the excessive purchase of meals or food ingredients, as well as the quantities wasted due to poor storage and transportation, and the quantities that may be damaged as a result of falling prices due to certain factors, policies or due to factors unsuitable environment. Therefore, all of the above must be mentioned and compared to the MFW, and the latter will inevitably be the least, and therefore the importance will be for the aforementioned.
Note: I do not see the use of the original research articles template appropriate here for this review study, because it is difficult to describe the materials and methods and the discussion paragraph is missing here in this study.
So, I suggest use the appropriate template for review studies such as:
Title - abstract- introduction - study objectives, then presenting the reference studies that achieve the objectives of the study - the conclusion through which we reach the most important useful results that achieve the objectives.
Author Response
Dear Reviewer 1,
Thank you very much for your comments! We appreciated a lot your valuable comments. It really helped us. We made corrections and necessary mentions.
Please see below.
Point 1: In Abstract:
- It should be indicated to the method of work used in the abstract, after the objective of the study, i.e. in line 14 approximately in this study.
Response 1: The method of work is now specified in the abstract section, as you suggested.
Point 2: Keywords:
- Include the term The Metabolic Food Waste [MFW] in the study keywords.
Response 2: The therm “Metabolic Food Waste [MFW]” is now included in the keywords, as you suggested.
Point 3: In introduction:
- Expand the introduction more by expanding more in the explanation of the concept of MFW and also inferring from recent references on the topic from several different countries around the world.
Response 3: The introduction has been now expanded, as you suggested, with more information and explanation of the concept of MFW. Other recent references on the topic are unavailable, as the only studies regarding the concept of Metabolic Food Waste are the ones already mentioned in the paper.
Point 4: In introduction:
- What was mentioned in lines 36-39 must be conveyed and used in conclusion or discussion (if any).
Response 4: The hypothesis presented in lines 36-39 (current lines 51-56) “We believe that the main reason for overconsumption and obesity is the lack of nutritional education of the consumers. Informing consumers about their physiological food and educating them in choosing healthy and sustainable food products can have a major impact in reducing the occurrence of food waste in the form of excessive consumption.” is now reiterated and expanded in subsection 3.2.4, as you suggested.
Point 5: In Materials and Methods:
- Research methods are very succinct.
Response 5: The research methods have been now explained in more details, as you suggested.
Point 6: In Materials and Methods:
- Expand more Explanation of the study methods according to the ideas put forward in the results sequentially.
Response 6: The study methods have been now further expanded and explained, and the “Materials and Methods” section was renamed “Study objectives” to better reflect its content, as you suggested.
Point 7: In Materials and Methods:
- Include the publication date of both references: Fischer’s and Garnett’s “Plates, pyramids, 47 planet” and the Barilla’s “Double Pyramid” in the body of the text on lines 47 and 48.
Response 7: The publication date for Fischer’s and Garnett’s “Plates, pyramids, and planet” and Barilla’s “Double Pyramid” has been now added in the body of text, as you suggested.
Point 8: In Results:
- Reconsider the way the references are placed in the end of the paragraphs. It means placing the references directly at the end of the reference idea and not stacking the references at the end of the paragraph (for example: lines 66 & 87.
Response 8: The references are stacked at the end of line 66 (current line number 100) because it is an enumeration of studies that simultaneously touch on several of the previously mentioned topics (food wasting behaviours, eating habits, consumer behaviour related to food etc.). Placing the references directly at the end of the reference idea means mentioning several references multiple times. At the end of line 87 (current line number 138) the references are stacked because of the shortness of the previous paragraph and the multitude of sources that sustain the data about prevalence of obesity in USA and Europe.
Point 9: In Results
- Line 116: The title of Table 1 does not match the content. Since it contains only Metabolic Food Waste but overweight and obesity are shown in Figure 1.
Response 9: The title of Table 1 is “Metabolic Food Waste [MFW (tons of food)] associated with overweight and obesity – authors selection after [19]” and the table presents the volume of Metabolic Food Waste(tons of food) correlated to the degree of obesity and overweight population in each of the region. The obese and overweight population of Europe (EU) corresponds to 39,201,410,847 tons of food wasted from overconsumption, therefore Metabolic Food Waste.
Point 10: In Results
- Line 198: Revise paragraph 3.2. 4. Are all of what is mentioned in it the authorship of the authors of this study? Because there is only one reference at the end of the paragraph in line 226. Or that everything mentioned in the paragraph was taken from the same reference, and this (if it is) needs to be reviewed and other supporting references added.
Response 10: The first paragraph of subsection 3.2.4 presents general information regarding the trend of dieting in the Western developed world. The second paragraph of subsection 3.2.4 is based on the Report of The EAT-Lancet Commission from 2019, a commission consisting of 37 world-leading scientists from 16 countries from various scientific disciplines related to food and human health. Although the approach of the commission regarding transparency of data and the methodology was criticised by Francisco J Zagmutt and colleagues (DOI: https://doi.org/10.1016/S0140-6736(19)31903-8) the authors of the Report provided an answer clarifying the issues raised by Francisco J Zagmutt (DOI: https://doi.org/10.1016/S0140-6736(19)31910-5).
Point 11: In Results
- The study focuses on the excessive amount of Metabolic Food Waste (MFW), but it is better to mention the quantities of food wasted due to the excessive purchase of meals or food ingredients, as well as the quantities wasted due to poor storage and transportation, and the quantities that may be damaged as a result of falling prices due to certain factors, policies or due to factors unsuitable environment. Therefore, all of the above must be mentioned and compared to the MFW, and the latter will inevitably be the least, and therefore the importance will be for the aforementioned.
Response 11: The paper focuses on the amount of Metabolic Food Waste (MFW) because it is a form of food waste not considered previously. The food waste due to the excessive purchase of meals or food ingredients, as well as the quantities wasted due to poor storage and transportation, and the quantities that may be damaged as a result of falling prices due to certain factors, policies or due to factors unsuitable environment are all previously researched in an extensive manner. Metabolic Food Waste is a concept that not only considers the amount of food that is wasted due to overconsumption, but also the negative effects of said overconsumption on human health, the strain it puts on the medical system due to health issues related to obesity and overweight, and not to be forgotten, the environmental impact of this form of food waste. Yes, compared to all other forms of food waste, Metabolic Food Waste may have the least importance in terms of quantity of food wasted and environmental damage, but nevertheless, it is a form of food waste and should be addressed as extensively as any other. This is the reason the current review focuses solely on Metabolic Food Waste. It is not a comparative analysis of different types of food waste.
Point 12: Note: I do not see the use of the original research articles template appropriate here for this review study, because it is difficult to describe the materials and methods and the discussion paragraph is missing here in this study.
So, I suggest use the appropriate template for review studies such as:
Title - abstract- introduction - study objectives, then presenting the reference studies that achieve the objectives of the study - the conclusion through which we reach the most important useful results that achieve the objectives.
Response 12: The template of the paper was modified now, to be more appropriate for a review study, as you suggested.

Reviewer 2 Report
The short review "Metabolic Food Waste as Food Insecurity Factor – Causes and 2 Preventions" is rather too short in nature with no much communication.
The author of this article needs to be more intentional about their message and the technical content of the piece.
The introductory section if floating as it has no technical drive to sustain the rest section.
The material and method section is not speaking to the content therein
The result and method section also is not in alignment to the tittle
The work needs scientific beef up as well as professional English editing service
Conclude the article in more scientific and thorough manner
References should be checked for some missing contents
Author Response
Dear Reviewer 2,
Thank you very much for your comments! It really helped us and we appreciated a lot your valuable comments.
So, we made corrections and necessary mentions.
Please see below.
Point 1: The short review "Metabolic Food Waste as Food Insecurity Factor – Causes and Preventions" is rather too short in nature with no much communication.
Response 1: The review has been extended and new information added to it, as you suggested.
Point 2: The author of this article needs to be more intentional about their message and the technical content of the piece.
Response 2: We have completed the manuscript with complementary information, through which we have improved the message and the technical content, as you suggested.
Point 3: The introductory section if floating as it has no technical drive to sustain the rest section.
Response 3: The introductory section has been completed with more information related to the article, as you suggested, with the purpose to present the intention to present the results of this research.
Point 4: The material and method section is not speaking to the content therein
Response 4: The “Materials and Methods” section has been changed, the name of “Study objectives” being more suited for it, and new content has been added, as you suggested.
Point 5: The result and method section also is not in alignment to the tittle
Response 5: The section has been restructured to better reflect the objectives of the review, as you suggested.
Point 6: The work needs scientific beef up as well as professional English editing service
Response 6: We have improved the scientific quality of the manuscript, as you can see from the new version of it, and we have turned to a professional translator for proofreading, as you have suggested.
Point 7: Conclude the article in more scientific and thorough manner
Response 7: The paperwork presents a topic that can be appreciated as personal by those who find themselves in a situation of obesity and overweight, and this is why we have formulated the conclusion in a way that could prevent them, trying not to offend anyone. At the same time, we have detailed the conclusions, as you suggested.
Point 8: References should be checked for some missing contents
Response 8: References have been updated, as you suggested.

Round 2
Reviewer 1 Report
Thank you for the good and clear scientific answer to the comments, and thank you for correcting and adding what was required in the previous report.